# Amplicon Sequencing Reveals Novel Fungal Species Responsible for a Controversial Tea Disease

**DOI:** 10.3390/jof8080782

**Published:** 2022-07-26

**Authors:** Yunqiang He, Yan Li, Yulin Song, Xingming Hu, Jinbo Liang, Karim Shafik, Dejiang Ni, Wenxing Xu

**Affiliations:** 1Hubei Hongshan Laboratory, Wuhan 430070, China; heyunqiang@webmail.hzau.edu.cn (Y.H.); ly632973748@163.com (Y.L.); nidj@mail.hzau.edu.cn (D.N.); 2Key Laboratory of Horticultural Plant Biology (Ministry of Education), Wuhan 430070, China; karim.awad@alexu.edu.eg; 3College of Plant Science and Technology, Huazhong Agricultural University, Wuhan 430070, China; 4Key Lab of Plant Pathology of Hubei Province, Wuhan 430070, China; 5Tea Industry Office, Agriculture and Rural Bureau of Zigui County, Yichang 443699, China; songyulin222@163.com; 6Agriculture and Rural Bureau of Enshi, Enshi 445015, China; xiangjun@tricaas.com; 7Academy of Agricultural Sciences of Enshi, Enshi 445002, China; esljb2022@163.com; 8Department of Plant Pathology, Faculty of Agriculture, Alexandria University, Alexandria 21526, Egypt

**Keywords:** high-throughput sequencing, amplicon sequencing, etiological identification, fungal identification, *Didymella*, two new taxa

## Abstract

Amplicon sequencing is a powerful tool for analyzing the fungal composition inside plants, whereas its application for the identification of etiology for plant diseases remains undetermined. Here, we utilize this strategy to clarify the etiology responsible for tea leaf brown-black spot disease (LBSD), a noticeable disease infecting tea plants etiology that remains controversial. Based on the ITS-based amplicon sequencing analysis, *Didymella* species were identified as separate from *Pestalotiopsis* spp. and *Cercospora* sp., which are concluded as the etiological agents. This was further confirmed by the fungal isolation and their specific pathogenicity on diverse tea varieties. Based on the morphologies and phylogenetic analysis constructed with multi-loci (ITS, LSU, *tub2**,* and *rpb2*), two novel *Didymella* species—tentatively named *D. theae* and *D. theifolia* as reference to their host plants—were proposed and characterized. Here, we present an integrated approach of ITS-based amplicon sequencing in combination with fungal isolation and fulfillment of Koch’s postulates for etiological identification of tea plant disease, revealing new etiology for LBSD. This contributes useful information for further etiological identification of plant disease based on amplicon sequencing, as well as understanding, prevention, and management of this economically important disease.

## 1. Introduction

The next-generation sequencing (NGS, alternatively termed second-generation sequencing) method was developed in the second half of the 2000s and marked the beginning of high-throughput sequencing (HTS) analyses of fungal communities [1]. With this approach, the internal transcribed spacer (ITS) regions, short (<500 bp) intergenic regions of the nuclear ribosomal RNA (rRNA) operon of fungi, are amplified using the polymerase chain reaction (PCR) technique and analyzed by NGS technology to identify and differentiate multiple microbial species from complicated samples. Since sequencing of the ITS region is a critical step in HTS studies on fungal communities, ITS-based amplicon sequencing (or amplicon metagenomic sequencing) is alternatively termed for HTS analyses, which have been widely applied for analyses of overall fungal diversity, saprotrophic fungi, mycorrhizal fungi, foliar endophytes, aquatic fungi, and human-associated fungi [1]. However, ITS-based amplicon sequencing in mycopathology has somewhat lagged behind other fields of mycology despite the potential usefulness of this approach for surveillance [1]. 

Because of its health benefits, tea represents an integral part of human routines and ranks as one of the three main popular non-alcoholic beverages worldwide. Tea tree (*Camellia sinensis* (L.) O. Kuntze) is one of the most important cash crops and is widely cultivated in more than 1000 counties and 20 provinces in China, the country with the leading tea yield in the world, and it plays an important role in the local economy and people’s daily life [2]. Several years ago, a noticeable disease infecting tea plants, named tea leaf brown-black spot disease (LBSD) [3] emerged in various tea-cultivating regions in China due to increased rainfall. This disease is only observed on the tender leaves of some specific tea varieties, e.g., Fuyun No. 1 and No.6; Taicha No. 2; and unauthorized varieties, like those locally cultivated in Zigui County (e.g., var. Ziguibendizao) and Enshi autonomous prefecture (Enshitaizicha), Hubei Province. This disease is locally known in Chinese as “chixingbing”, which means red (chi) star (xing) disease (bing). In the early stages of LBSD, tiny (approximately 1–2 mm in diameter) light-yellowish spots turned into blackish- or reddish-brown spots. The late stage of infection develops on tender leaves, affecting tea quality and causing it to taste bitter and astringent [3]. In previous studies, the etiological agents responsible for LBSD were identified as *Pestalotiopsis theae*, *P. camelliae,* and *P. clavispora* [3]. We still cannot completely exclude other potential fungi as being responsible for LBSD, since the previous identification strongly depended on the fungal isolation, and some pathogens were easily omitted if they were uncultured or difficult to grow on potato dextrose agar (PDA) medium. Moreover, *Phoma segeticola var. camelliae* was also reported in association with this disease in China’s Guizhou Province [4]. Therefore, LBSD etiology remains controversial and calls for a *de novo* identification of LBSD by a full-scale approach to obtain insight into the fungal pathogens inside the diseased plants. 

Here, we reveal *Didymella* spp., which was determined to be responsible for LBSD by using ITS-based amplicon sequencing in combination with fungal isolation and fulfillment of Koch’s postulates. This study provides an integrated approach to identifying the fungal etiology responsible for plant disease, and two novel *Didymella* species are proposed and characterized.

## 2. Materials and Methods

### 2.1. Sample Collection

In August 2020, tender tea leaf samples (termed SZX, JYC, and WJT—referring to the town or village Shazhenxi, Jieyacun, and Wujiatai, where the samples were collected, respectively) showing typical symptoms of brown-black spot disease (1–2 mm in diameter) (Figure 1A) were collected from var. Ziguibendizao in Daozuopu, and Jieyacun Villages in Shazhenxi Town, Zigui County, Yichang City, and var. Enshitaizicha in Wujiatai Town, Xuan’en County, Enshi Autonomous Prefecture, Hubei Province, China (Appendix A). A sample NC showing no symptoms was simultaneously collected from the same variety as SZX from Daozuopu village. The leaves of WH (referring to the sampling City Wuhan) sample showing leaf blight symptoms were collected from Hongshan District, Wuhan City, Hubei Province, China (Appendix A). At least three leaves for each sample were collected for further analysis.

### 2.2. ITS-Based Amplicon Sequencing and Bioinformatic Analysis

Collected leaf samples were frozen in liquid nitrogen and then subjected to ITS-based amplicon sequencing by Beijing Novogene Technology Company Limited.

Total genomic DNA was separately extracted from tea leaf samples using a CTAB-based method [5]; concentration was assured using electrophoresis on 1% agarose gel, and the solution was diluted to the final concentration of 1 ng/µL and subjected to PCR amplification using the primer pairs, ITS5 and ITS2 [6]. Amplification conditions began with an initial denaturation at 98 °C for 1 min, followed by 30 cycles of 98 °C for 10 s, 50 °C for 30 s, and 72 °C for 30 s, and terminated at 72 °C for 5 min. The PCR products were purified with Qiagen Gel Extraction Kit (Qiagen, Germany) after being fractionated by electrophoresis on agarose gel (2%, w/v) and generated in a sequencing library with TruSeq® DNA PCR-Free Sample Preparation Kit (Illumina, San Diego, CA, USA). The DNA was sequenced on an Illumina NovaSeq platform after qualified on the Qubit@ 2.0 Fluorometer (Thermo Scientific, Waltham, MA, USA) and Agilent Bioanalyzer 2100 system, resulting in approximately 250 bp paired-end reads.

The resulting reads were truncated by removing the barcode and primer sequence, merged using FLASH (V1.2.7, http://ccb.jhu.edu/software/FLASH/, accessed on 1 November 2020) [7], and subjected to a quality control process under specific filtering conditions [8] with QIIME (V1.9.1, http://qiime.org/scripts/split_libraries_fastq.html, accessed on 1 November 2020) [9]. The chimera sequences were removed after being aligned with the sequences deposited in the reference database (Silva database, https://www.arb-silva.de/, accessed on 1 November 2020) using the UCHIME algorithm (UCHIME http://www.drive5.com/usearch/manual/uchime_algo.html, accessed on 1 November 2020) [10,11]. 

Sequence analysis was conducted using Uparse software (Uparse v7.0.1001, http://drive5.com/uparse/, accessed on 1 November 2020) [12], and the sequences were assigned to the same operational taxonomic units (OTUs) when their identity was higher than 97%. The taxonomic situation was annotated based on the Mothur package with the Silva Database (http://www.arb-silva.de/, accessed on 1 November 2020) [13]. The phylogenetic relationships of different OTUs were analyzed based on the multiple sequence alignments using MUSCLE software (Version 3.8.31, http://www.drive5.com/muscle/, accessed on 1 November 2020) [14], and the abundance was calculated based on the sequence number corresponding to the total sequences within each sample. 

### 2.3. Fungal Isolation

Fungal isolation and purification were performed as previously described [15]. At least four small pieces of tissues (25 mm^2^) were excised from each tea leaf after it was surface-sterilized with 5% NaClO for 3–5 min and 75% ethanol for 15 s and incubated at 25 °C in darkness for 3 to 5 days on PDA medium (20% diced potatoes, 2% glucose, and 1.5% agar) for fungal colony formation. The developed colonies were further purified and stored in 25% glycerol at −70 °C until use. For the newly identified fungal species, their ex-type living cultures and type specimens were deposited in the China General Microbiological Culture Collection Centre (CGMCC) and the Mycological Herbarium, Institute of Microbiology, Chinese Academy of Sciences (HMAS), Beijing, China, respectively.

### 2.4. Virulence and Pathogenicity Tests

Virulence tests with mycelia were carried out as previously described [3,15]. The mycelial plugs were taken from the margin of 3-day-old colonies and inoculated on attached or detached tea leaves (vars. Enshitaizicha, Ziguibendizao, E’cha No. 1, and Fudingdabai) under wounded conditions (three wounds were made using a tiny needle in the middle of each leaf) after the leaves were washed in sterilized water and then left for air-drying. The inoculated detached leaves were incubated at 25 °C in a relative humidity of approximately 90% and with a photoperiod of 12 h. Uncolonized PDA plugs were involved in parallel as negative controls. Five leaves were used for each treatment, and the symptoms were observed daily. 

A pathogenicity test with conidia was implemented on the attached leaves of tea plants (var. Enshitaizicha) in Wuhan, Hubei province, China, in the early spring (i.e., at the end of March 2022), as previously described [16]. The conidial suspensions (10^5^ conidia/mL) diluted in PD medium were sprayed on the newly developed buds after they were mechanically ground with carborundum particles. The PD solution without conidia was involved in parallel as a control, and ten buds were inoculated for each treatment. The inoculated and uninoculated leaves were covered with plastic membrane for 24 h, kept in natural conditions for symptom development, and observed every 3 days. 

### 2.5. DNA Extraction, Amplification, and Sequencing

Purified fungal isolates were separately cultured on PDA media at 25 °C in darkness for 5 days and subjected to DNA extraction using the silica-spin-column-based method, as previously described [17]. The extracted DNA was used for amplifying the internal transcribed spacer (ITS), the partial large subunit 28S nrDNA region (LSU), *β**-tubulin* (*tub2*), and the second-largest RNA polymerase II subunit regions (*rpb2*) using PCR using 2×GimiTaq PCR Mix (Gimmico, Wuhan, China) with the primer pairs of ITS1 and ITS4 [6] for ITS regions, LR0R [18] and LR7 [19] for LSU regions, TUB2Fw and TUB4Rd [20] for *tub2* regions, and RPB2-5F2 and fRPB2-7cR [21] for *rpb2* regions based on the reported PCR programs. All amplicons were purified and sequenced by Beijing Tsingke Biological Technology Company Limited (Wuhan, China).

### 2.6. Phylogenetic Analysis

The obtained ITS sequences were aligned with the sequences deposited in the NCBI (http://www.ncbi.nlm.nih.gov) database using the BLASTn program, and the phylogenetically related fungal isolates and their ITS, LSU, *tub2*, and *rpb2* sequences were downloaded for further phylogenetic analysis [22]. The obtained nucleotide sequences were trimmed to similar sizes using DNAMAN software (version 9.0.1.116), aligned by MAFFT (https://www.ebi.ac.uk/Tools/msa/mafft/, accessed on 1 March 2022), manually adjusted in MAGE7 [23] if necessary, and concatenated according to the order of ITS, LSU, *rpb2,* and *tub2* [24] using PhyloSuite (Version 1.2.2) [25]. The concatenated aligned dataset consisted of 2020 nucleotides and gaps (ITS = 1–458 bp; LSU = 459–1152 bp; *rpb2* = 1153–1747 bp; *tub2* = 1748–2020 bp). A maximum-likelihood (ML) phylogenetic tree was constructed with the aligned sequences using IQ-TREE [26] under the GTR model for 1000 ultrafast bootstraps [27] with a Shimodaira–Hasegawa-like approximate likelihood ratio test [28]. FigTree v1.4.3 [29] was used to view the phylogenetic tree. 

MrBayes 3.2.6 [30] was used to analyze the interspecific relationship using Bayesian inference (BI). The best partitioning scheme and evolutionary models for 4 predefined partitions were selected using PartitionFinder2 [31] using a greedy algorithm and BIC criterion. The best nucleotide substitution models recommended by PartitionFinder2 for Bayesian analysis were SYM+G for ITS, K80+I+G for LSU, and SYM+G for both *rpb2* and *tub2*. The Markov chain Monte Carlo (MCMC) algorithm operated for 2,000,000 generations until the average standard deviation of frequency was less than 0.01. The trees summarized after abandoning the aging samples were obtained, and the posterior probabilities (PP) of each branch were calculated. Sequences generated in this study were deposited in GenBank (Appendix A), and their taxonomic information was deposited in MycoBank [32].

### 2.7. Morphological Assessments and Growth Rate

Mycelial plugs of each isolate were excised from the margins of 3-day-old colonies and cultured on PDA at 25 °C with a photoperiod of 14 d until pycnidium formation. The growth of each colony was observed and measured daily. Fungal conidia were diluted to a low concentration and observed under an optical microscope (Olympus microscope BX63, Olympus Corporation, Tokyo, Japan). Fifty conidia were randomly selected to be measured and then photographed with the SPOT program (Version 4.6, Diagnostic Instruments, Inc., Sterling Heights, MI, USA). Pycnidia were promptly formed by culturing the mycelial discs (5 mm diameter) on alfalfa stems to produce a fungal fruiting body at 25 °C for 14 d until conidia were produced, then sectioned and observed under an optical microscope (Olympus microscope BX63, Olympus Corporation, Tokyo, Japan). Alfalfa stems were utilized instead of tea leaves since the fungal fruiting bodies were easily produced on these tissues in a short time [33,34], whereas they had difficulty forming on the detached tea leaves until the leaves turned to rot. The growth rate of each isolate was estimated with colony semidiameter dividing the cultured days after subtracting the initial disk size. Three replicates of each isolate were used.

## 3. Results 

### 3.1. ITS-Based Amplicon Sequencing Reveals the Candidate Etiology

Three naturally infected tender leaf samples (SZX, JYC, and WJT) showing typical LBSD symptoms together with an asymptomatic leaf sample (NC) were collected from four different gardens in two different counties of Hubei province, China, frozen by liquid nitrogen, and then subjected to ITS-based amplicon sequencing. Four libraries (for SZX, JYC, WJT, and NC samples, respectively) were prepared from ITS-based amplicon and sequenced with the Solexa-Illumina platform. A totals of 66,187, 63,279, 62,994, and 64,043 effective tags were obtained for SZX, JYC, WJT, and NC, respectively, after removing low-quality reads. The average contigs were 234 to 242 bp in length (Appendix A). 

Bioinformatic analysis revealed over 402 fungal genera detected in the collected samples. Ascomycota fungi were the most abundant, with more than 76% incidence in each sample, followed by Basidiomycota fungi, with a proportion of less than 6% (Appendix A). Correlation analysis of the abundance of the genera with the samples revealed that the genera belonging to Ascomycota obviously accumulated in the systematic samples, while no genera belonging to Basidiomycota or others did (Appendix A). A phylogenetic tree constructed based on the ITS sequences of these Ascomycota fungi revealed 20 fungal species belonging to 20 genera, 16 families, 10 orders, and 7 classes of this phylum, while their total amount was similar in each sample (Figure 1B), supporting credible, unbiased deep sequencing and analysis. Among them, *Didymella* spp. were detected in all diseased samples at a considerably high abundance (no less than 1.00%, i.e., 5.27% in SZX, 4.32% in JYC, and 18.89% in WJT) but had a very low abundance (0.88%) in the NC sample. In contrast, *Cladosporium* sp. dominated in the asymptomatic leaves as well as in the diseased samples (Figure 1B); both *Cercospora* sp. and *Phaeosphaeria* sp. were detected in both WJT and JYC leaves at a considerably high abundance (4.54% and 2.81%, respectively) but not in SZX, and the remaining fungal species were detected in one or two samples at a low incidence (less than 0.01%) (Figure 1B). This depicts that *Didymella* spp. are the candidate etiological agents responsible for LBSD, since only these fungi could be detected in all the diseased samples at considerably high amounts. *Didymella* spp. were also detected in the control sample at a very low incidence, which might be the result of contamination. To confirm this, four tea leaf samples (WH) with no LBSD-like symptoms were collected for ITS-based amplicon sequencing, and as expected, no *Didymella* spp. were detected (Appendix A). 

### 3.2. Fungal Isolation Confirmed the ITS-Based Amplicon Sequencing Analysis

To confirm the fungal species detected by amplicon sequencing in the diseased tea samples, the collected leaves were subjected to fungal isolation. After single mycelium purification, 10, 19, 20, and 18 colonies were obtained from JYC, SZX, WJT, and NC samples, respectively. Of these, *Didymella* spp. were found to be abundantly detected with an incidence of approximately 20.0% in each symptomatic sample, while they were not detected in the healthy NC sample leaves (Table 1). In contrast, *Pestalotiopsis* spp. were only isolated from SZX and WJT samples (with incidences of 21.1% and 55.0%, respectively), *Colletotrichum* spp. were found only in JYC and SZX samples (20.0% and 21.1%, respectively); *Cercospora* spp. were found in the JYC sample (10.0%) as well as the remaining fungi from one or two samples (5.0% to 20.0%) (Table 1). 

### 3.3. Virulence Tests on Detached Tea Leaves Reveal Didymella spp. as the Candidate Etiological Agents

Three *Didymella* isolates (JYC-1-9, SZX-1-9, and WJT-2-3), together with *Pestalotiopsis* isolates (SZX-1-1, WJT-2-4, and WJT-1-1) and the only one obtained *Cercospora* isolate (JYC-1-2), were selected so that their virulence on three tea varieties (vars. Enshitaizicha, E’cha No. 1, and Fudingdabai) could be studied, since the two latter fungal species had been highly speculated to be the etiological agents. After inoculation on the tender leaves, over 5 days post-inoculation (dpi), all *Didymella* isolates induced many small brown spots (approximately 9–18 spots per leaf) on the leaves of Enshitaizicha, highly similar to the LBSD symptoms observed in the field, whereas very few or no spots were observed on the leaves of E’cha No. 1 (0–2) or Fudingdabai (0–1) (Figure 2A). The symptoms of the three tea varieties correspond to the diverse symptom severity observed in the field, since var. Enshitaizicha is sensitive to this disease, while both vars. E’cha No. 1 and Fudingdabai are not. In contrast, *Pestalotiopsis* induced significantly fewer spots (1–3 spots per leaf) on the leaves of both vars. Enshitaizicha and E’cha No. 1, and no spots on var. Fudingdabai; *Cercospora* isolates induced no lesions, or much smaller brown spots, on all the inoculated leaves (Figure 2A). 

Fungal reisolation from the diseased tissues neighboring the asymptomatic ones confirmed that the subisolates matched the inoculated ones with regards to their morphologies and ITS sequences (data not shown). These results support that *Didymella* spp., instead of *Pestalotiopsis* or *Cercospora* fungi, are the candidate etiological agents responsible for LBSD.

### 3.4. Phylogenetic Analyses

For further identification of the *Didymella* spp. related to LBSD, *Didymella* isolates were subjected to sequencing of their ITS, LSU, *rpb2*, and *tub2* regions. The obtained sequences, along with 25 representative *Didymella* species retrieved from the GenBank database based on BLASTn searches of their ITS sequences, were subjected to phylogenetic analysis. The phylogenetic tree reveals that ten isolates formed two unique clades with a high degree of support (1.00/98 and 0.96/96)—which were separated from known *Didymella* species, with the closest relationships being with *D. segeticola* and *D. bellidis*—and identified two novel species of *Didymella*, tentatively named *Didymella theae* W.X. Xu and Y.Q. He (six strains) and *Didymella theifolia* W.X. Xu & Y.Q. He (four strains), respectively (Figure 3) (see below), as referring to the definition and naming rules for a new fungal species as previously described [35].

### 3.5. Taxonomy

***Didymella theae*** W.X. Xu and Y.Q. He, sp. nov. (Figure 4)

MycoBank: MB 842446

**Etymology.** Named for the host plant, *C. sinensis* (L.) O. Ktze., from which the species was isolated. 

**Description.***Asexual morph* developed on alfalfa. *Conidiomata* pycnidial, spherical or oblate, single or fused, with a smooth surface, 57.3–366.3 × 53.0–328.3 μm, 1–2 orifices, and long papillae. *Pycnidial wall* pseudoparenchymatous, composed of 3–4 layers of rectangular cells, light brown in the early stage and dark brown in the later stage. *Conidiogenous cells* phialidic, hyaline, simple, smooth, flask-shaped to sometimes isodiametric, 3.7–6.7×3.5–5.2 μm. *Conidia* oval, transparent, smooth and without diaphragm, 3.6–5.0 × 2.0–3.0 μm (mean = 4.2 ± 0.4 μm × 2.5 ± 0.2 μm, n = 50), with 0–2 polar guttules at both ends. *Chlamydospores* absent. *S**exual morph* not observed.

**Culture characteristics.** Colonies on PDA medium, 67.5 mm diam after 7 d. Colony surface brown, margin regular, densely covered by white fluff, white aerial mycelia, reverse dark brown. Spherical light-yellow conidiomata formed on the mycelial surface as cultured over 14 days. 

**Materials examined.** China, Hubei Province, Yichang City, on diseased leaves of *C. sinensis* var. Enshitaizicha, 20 August 2020, W.X. Xu and Y.Q. He (holotype HMAS 351945, ex-holotype culture CGMCC 3.20886 = WJT-2-3); ibid., ex-isotype culture SZX-3-2, SZX-3-7, WJT-1-2, WJT-2-7, WJT-2-9.

**Notes.** In the multi-site phylogenetic tree, *D. theae* forms a branch with a high degree of support (1/98) and has the closest relationship with *D. segeticola* and *D. bellidis*. In morphology, the pycnidia of *D. theae* is larger than those of *D. segeticola* and *D. bellidis* (57.3–366.3 × 53.0–328.3 μm in *D. theae* vs. 90.0–105.0 × 75.0–95.0 μm in *D. segeticola*, 50.0–260.0 μm in *D. bellidis*). The conidia of *D. theae* are shorter than those of *D. segeticola* and *D. bellidis* (3.6–5.0 × 2.0–3.0 μm in *D. theae* vs. 4.5–7.0 × 2.5–4.0 μm in *D. segeticola*, 3.8–6.4 × 1.8–2.6 μm in *D. bellidis*) [24,36].

***Didymella theifolia*** W.X. Xu & Y.Q. He, sp. nov. (Figure 5).

**MycoBank:** MB 842458

**Etymology.** Named for the host plant, *C. sinensis* (L.) O. Ktze., from which the species was isolated. 

**Description.***Asexual morph* developed on alfalfa. *Conidiomata* pycnidial, spherical or oblate, single or fused, with a smooth surface, 65.3–329.6 × 44.0–299.6 μm, with a single papillate ostiolar neck. *Pycnidial wall* pseudoparenchymatous, composed of 3–4 layers of textura angularis, light brown in the early stage and dark brown in the later stage. *Conidiogenous cells* phialidic, hyaline, simple, smooth, flask-shaped to sometimes isodiametric, 4.5–7.5 × 3.2–5.7 μm. *Conidia* oval, transparent, smooth and without diaphragm, 3.7–5.5 × 2.1–2.8 μm (mean = 4.6 ± 0.4 μm × 2.5 ± 0.2 μm, n = 50), with 0–2 polar guttules at each end. *Chlamydospores* absent. *Sexual morph* not observed.

**Culture characteristics.** Colonies on PDA medium, 77.1 mm diam after 7 d. Colony center yellowish brown, margin regular, densely covered by white fluff, white aerial mycelia, reverse light brown. Spherical light-yellow spherical-shaped conidiomata formed on the surface of the colony over 14 days. 

Materials examined. China, Hubei Province, Yichang City, on diseased leaves of *C. sinensis* var. Fudingdabai, 20 August 2020, W.X. Xu and Y.Q. He (holotype HMAS 351946, ex-holotype culture CGMCC 3.20887 = JYC-1-9); ibid., ex-isotype culture JYC-1-6, SZX-1-9, SZX-1-10.

**Notes.** In the multi-site phylogenetic tree, *D. theifolia* form a branch with a high degree of support (0.96/96) and have the closest relationship with *D. segeticola* and *D. bellidis*. In morphology, *D. theifolia* is different from *D. segeticola* and *D. bellidis* in its larger pycnidia (65.3–329.6 × 44.0–299.6 μm in *D. theifolia* vs. 90.0–105.0 × 75.0–95.0 μm in *D. segeticola*, 50.0–260.0 μm in *D. bellidis*), but the conidia of *D. theifolia* are shorter than those of *D. segeticola* and *D. bellidis* (3.7–5.5 × 2.1–2.9 μm in *D. theifolia* vs. 4.5–7.0 × 2.5–4.0 μm in *D. segeticola*, 3.8–6.4 × 1.8–2.6 μm in *D. bellidis*) [24,36].

### 3.6. Fulfillment of Koch’s Postulates Confirms Both Didymella Species as the Etiological Agents

To further confirm both *Didymella* species responsible for LBSD, the conidia of six isolates (JYC-1-9, SZX-1-9, SZX-1-10, WJT-1-2, WJT-2-3, and WJT-2-9) of both *Didymella* species were further sprayed onto the attached tender leaves or buds of var. Enshitaizicha under wounded or unwounded conditions in the field in March. From 12–30 dpi, many small brown spots were gradually observed on the inoculated leaves under both conditions, and the symptoms matched the original symptoms observed in the field, while no symptoms were observed on the control leaves sprayed with sterilized water containing no fungal conidia (Figure 6A). 

Fungal reisolation conducted from the diseased tissues neighboring the asymptomatic tissues revealed that the subisolates matched the inoculated tissues with regards to their morphologies and ITS sequences (data not shown). These results support that *Didymella* spp. are the etiological agents responsible for LBSD.

### 3.7. Virulence and Growth Rate of the Didymella Isolates

*Didymella* isolates were also subjected to an assessment of their virulence under wounded conditions on the detached tea leaves of var. Ziguibendizao, another tea variety sensitive to LBSD, similarly to Enshitaizicha and E’cha No.1. The results showed a diverse virulence for the isolates within each species on both varieties, resulting in the lesions ranging from 7.0 to 11.5 mm on the leaves of var. Ziguibendizao and 13.2 to 13.9 mm on the leaves of E’cha No.1 for *D. theae* isolates and from 7.7 to 12.2 mm on the leaves of var. Ziguibendizao and from 12.4 to 15.0 mm on the leaves of E’cha No.1 for *D. theifolia* isolates, with a significant difference between some of them (Figure 6B,C). No correlation was found between the virulence and the fungal species.

The growth rates of these isolates were accessed as cultured on PDA media at 25 °C in darkness. The results showed that they had diverse growth rates for the isolates within each species, i.e., ranging from 8.8 to 10.2 mm/d for *D. theae* isolates and from 8.5 to 10.3 mm/d for *D. theifolia* isolates, with a significant difference between some of them (Figure 6D). No correlation was found between the growth rates and the species.

## 4. Discussion

Amplicon sequencing has been increasingly applied to characterize endophytic and pathogenetic fungal communities in plants [37,38,39]; it remains undetermined whether it can identify the etiological agents of plant diseases, although it has been utilized to detect bacterial and fungal organisms infecting soybean leaf samples [40]. In this study, the amplicon sequencing approach, together with fungal isolation and fulfillment of Koch’s postulates, was integrated for the identification of the etiological agents responsible for LBSD in tea plants. The inclusivity and sensitivity of the amplicon sequencing support that it is a powerful approach in combination with traditional fungal isolation and pathogenetic tests for identifying the etiological agents of plant disease. 

Here, LBSD was chosen for the etiological identification considering that its etiology remained controversial in recent years because more than one fungal species was highly isolated from the symptomatic leaves showing similar symptoms. Based on ITS-based amplicon sequencing and fungal purification, *Didymella* fungi are found to be the etiological agents responsible for LBSD since they had been abundantly detected and isolated from all diseased leaves while not from healthy samples. This was further confirmed by the pathogenicity tests on tea leaves with *Didymella* spp. by fulfilling Koch’s postulates. Additionally, *Didymella* spp. showed specific pathogenicity depending on the tea varieties, inducing more numerous and more apparent small brown spots on the leaves of Enshitaizicha and Ziguibendizao than on those of E’cha No. 1 and Fudingdabai (Figure 2 and Figure 6), which highly resembles the situation observed in the field, since LBSD is only observed on these specific tea varieties [3]. Although *Pestalotiopsis* spp. also caused small brown spots on the inoculated tea leaves, similar to the LBSD symptoms, they induced much less amount and less visualized spots on the leaves of var. Enshitaizicha in compared with *Didymella* spp. More importantly, *Pestalotiopsis* spp. caused the small brown spots independent of the varieties, suggesting that they are most likely not the causal agent of LBSD but simply cause similar symptoms. Additionally, *Cercospora* sp. was also excluded as the causal agent of LBSD, as it was only detected on JYC and WJT samples by amplicon sequencing and showed no virulence on any of the tested tea varieties, although it has been proposed to cause brown round spot disease, showing symptoms highly similar to LBSD [41]; *Cladosporium* sp. was highly detected in all symptomatic samples as well as in the asymptomatic samples, suggesting it is most likely an endophytic fungus infecting tea plants.

To confirm the result of amplicon sequencing, fungal isolation was performed on the collected tea leaves. It is noteworthy that the proportion of *Didymella* spp. is almost the same in the three samples as indicated by traditionally isolated manipulation, i.e., 20% for JYC and WJT and 21.1% for SZX, which differed from those illustrated by the amplicon sequencing (i.e., 5.27% for SZX and 4.32% for JYC), except for WJT (18.89%, similar to the isolation proportion) (Table 1 and Figure 1B). Additionally, *Pestalotiopsis* spp. and *Colletotrichum* spp. were highly isolated from both symptomatic leaves, while no data related to these fungi were obtained by the amplicon sequencing. In contrast, *Cladosporium* sp. was abundant in presence in all the tea leaves revealed by the amplicon sequencing but failed to be successfully isolated. Moreover, the fungal community (approximately 20 genera) revealed by bioinformatic analysis was obviously higher than that revealed by the traditional isolation strategy (approximately 11 genera). All these results suggest that amplicon sequencing is highly more sensitive than the traditional isolation approach, but it still has a range of pitfalls and potential biases, since the selection of genetic markers, primers, amplification conditions, and data processing and analysis affect the approach accuracy [1,42]; and it still requires further support from other approaches, e.g., fungal isolation and pathogenicity tests, for etiological identification. Moreover, the ITS region provides insufficient resolution for species-level assignment for several groups of important plant pathogens [43], and the candidate fungi require further identification in combination with other loci for species determination [1].

*Didymella* belong to Didymellaceae, which was established by De Gruyter, et al. [44] and includes *Ascochyta*, *Didymella*, *Phoma*, and other allied phoma-like genera. In previous studies, Didymellaceae species were identified mainly according to their morphologies and plant–host relationships, which are limited in species determination [24,45]. In recent years, phylogenetic analysis based on multi-loci has been developed for taxon analysis in *Phoma* and phoma-like genera [45,46], and the multi-loci (ITS, LSU, *rpb2*, and *tub2*) provide a solid phylogenetic pillar for the determination of Didymellaceae taxa [22,47,48,49,50]. Here, we utilized these multi-loci for phylogenetic analysis, revealing that these obtained isolates associated with LBSD symptoms of tea plants belong to two novel species, tentatively named *D. theae* and *D. theifolia* to refer to their host plants, with the suffix “–*folia*” used for a distinction from the former. In the phylogenetic tree, *D. theae* and *D. theifolia* has the closest relationship with *D. segeticola* and *D. bellidis*, while the former showed some difference in the morphologies as compared to the latter, with larger pycnidia and shorter conidia [24,36].

Collectively, amplicon sequencing is a powerful approach for analyzing the fungal composition inside plants, whereas it remains unexplored for the identification of etiological agents for plant diseases. Here, we applied ITS-based amplicon sequencing in combination with fungal isolation and fulfillment of Koch’s postulates to identify the etiological agents responsible for LBSD of tea plants, which provides solid evidence indicating that *D. theae* and *D. theifolia*, two novel proposed and characterized *Didymella* species, are responsible for this disease. This contributes useful information for further etiological identification of plant disease based on amplicon sequencing, as well as understanding, prevention, and management of this economically important disease.

## Figures and Tables

**Figure 1 jof-08-00782-f001:**
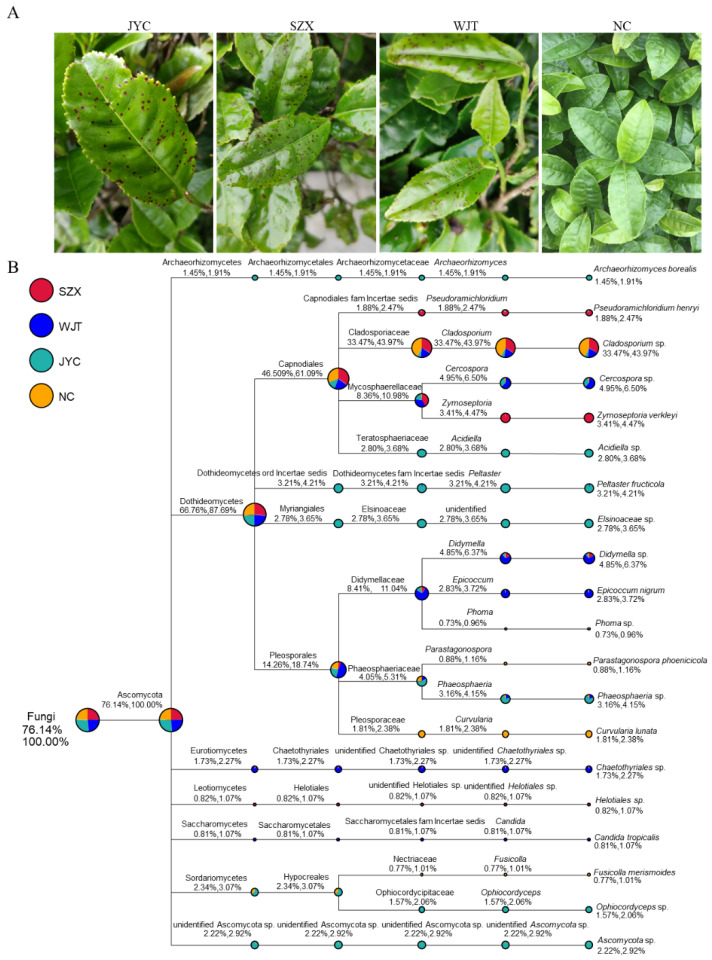
Sample symptoms and phylogenetic tree of Ascomycota fungi revealed by amplicon sequencing in samples SZX, WJT, JYC, and NC in the diverse taxonomic situation. (**A**) The symptoms of the collected tea leaves showing brown-black spot disease (LBSD) collected from three different regions. (**B**) The phylogenetic tree for twenty fungal species with leading relative abundance in each sample selected and involved in the analysis. The circles with different colors on the upper-left position refer to the diverse samples; the circles in different nodes indicate different taxonomic levels, with varied colors and sizes referring to the composition and relative abundance, respectively. The two numbers below the category name represent the proportion of their relative abundance accounting for all the fungal categories and Ascomycota, respectively.

**Figure 2 jof-08-00782-f002:**
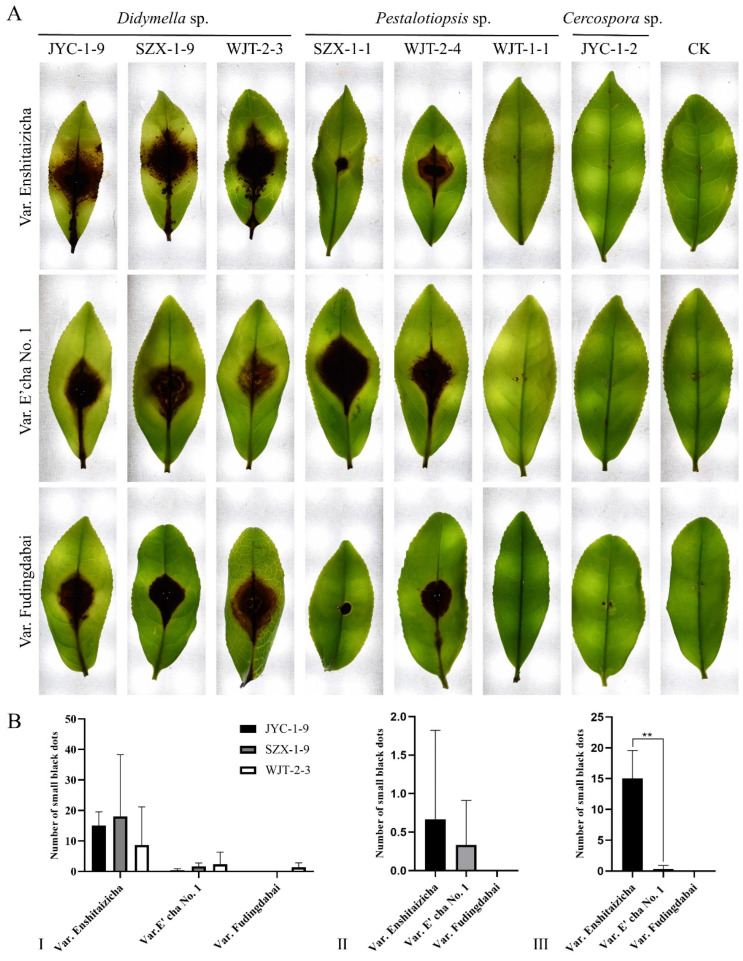
Pathogenicity analysis of *Didymella*, *Pestalotiopsis,* and *Cercospora* isolates on the tea leaves of vars. Enshitaizicha, E’cha No. 1, and Fudingdabai. (**A**) The representative leaf symptoms of the varieties induced by the involved isolates, indicated on the left and on the top of the panels, respectively. The pathogenicity test was conducted at 25 °C and 90% relative humidity and photographed at 5 days post-inoculation (dpi). (**B**) The small brown spot numbers on the leaves induced by the *Didymella* (I) and *Pestalotiopsis* isolates (Ⅱ) and the statistics analysis for the treatment of *Didymella* JYC-1-9 (Ⅲ). The data were analyzed by SPSS Statistics 19.0 (International Business Machines Corporation, Armonk, NY, USA) using one-way ANOVA, and the mean values were compared with a Tukey test, *p* = 0.01. The asterisks (**) on error bars indicate significant differences at *p* = 0.01.

**Figure 3 jof-08-00782-f003:**
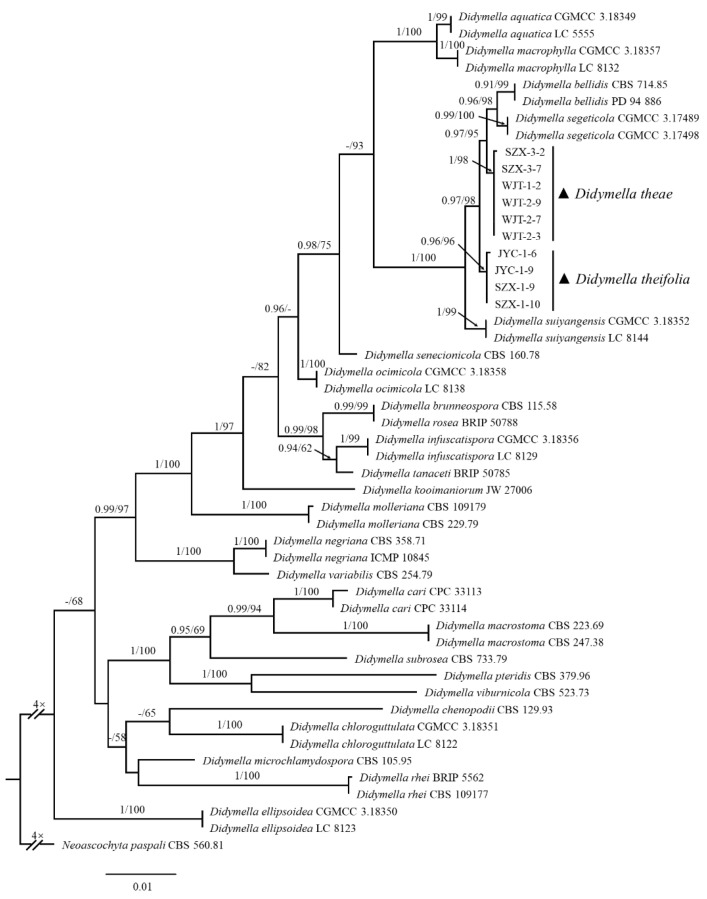
Phylogenetic tree generated by Bayesian inference (BI) and maximum likelihood (ML) based on combined ITS, LSU, *rpb2,* and *tub2* sequences of *Didymella theae*, *D. theifolia,* and the 25 species that have the closest phylogenetic relationship with them. *Neoascochyta paspali* (CBS 560.81) was selected as the outgroup. The ML tree was used to show the phylogenetic relationships of isolates and reference strains. Bayesian posterior probability (PP ≥ 0.90) and ML bootstrap support values (ML ≥ 50%) are shown at the nodes (PP/ML).

**Figure 4 jof-08-00782-f004:**
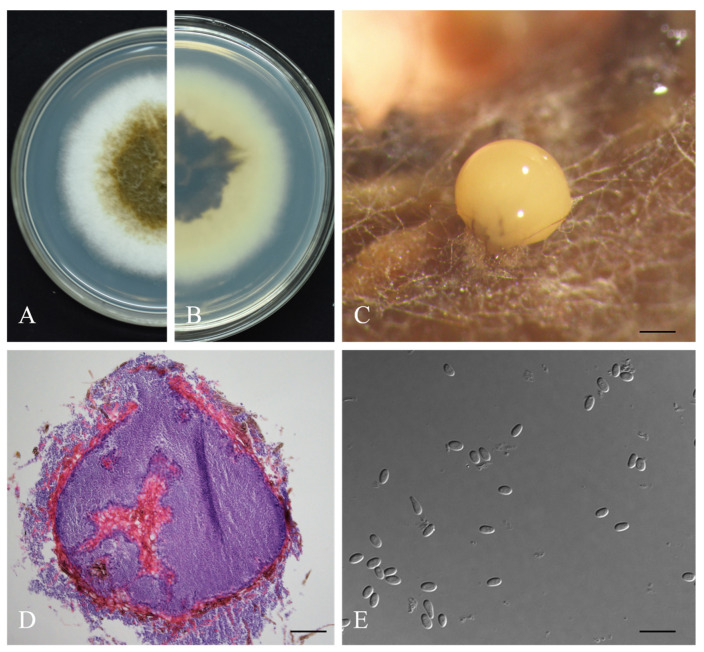
The morphologies of *Didymella theae*. (**A**,**B**) Front and back views of colonies cultured on PDA for 7 days, respectively. (**C**) Conidioma. (**D**) Section view of pycnidia. (**E**) Conidia. Scale bars indicate 200, 50, and 10 μm in panels (**C**), (**D**) and (**E**) respectively.

**Figure 5 jof-08-00782-f005:**
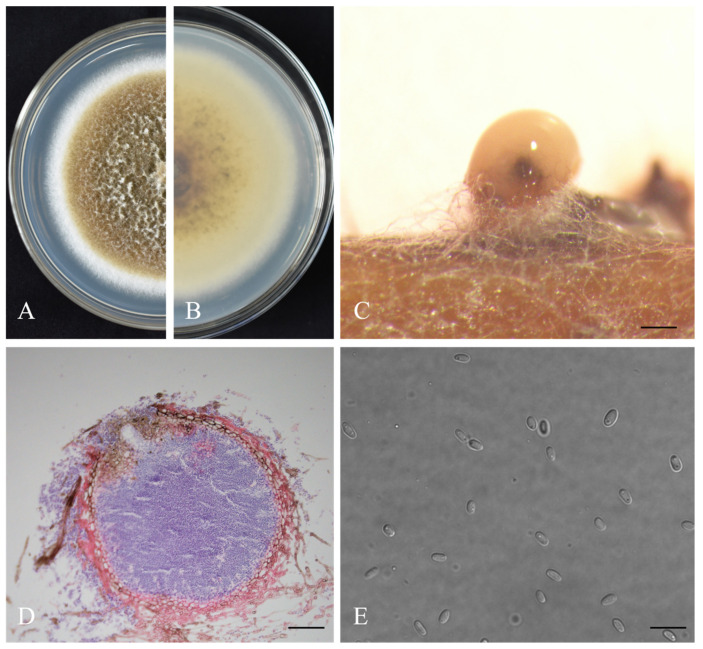
The morphologies of *Didymella theifolia*. (**A**,**B**) Front and back views of colonies cultured on PDA for 7 days, respectively. (**C**) Conidioma. (**D**) Section view of pycnidia. (**E**) Conidia. Scale bars indicate 200, 50, and 10 μm in panels (**C**), (**D**), and (**E**), respectively.

**Figure 6 jof-08-00782-f006:**
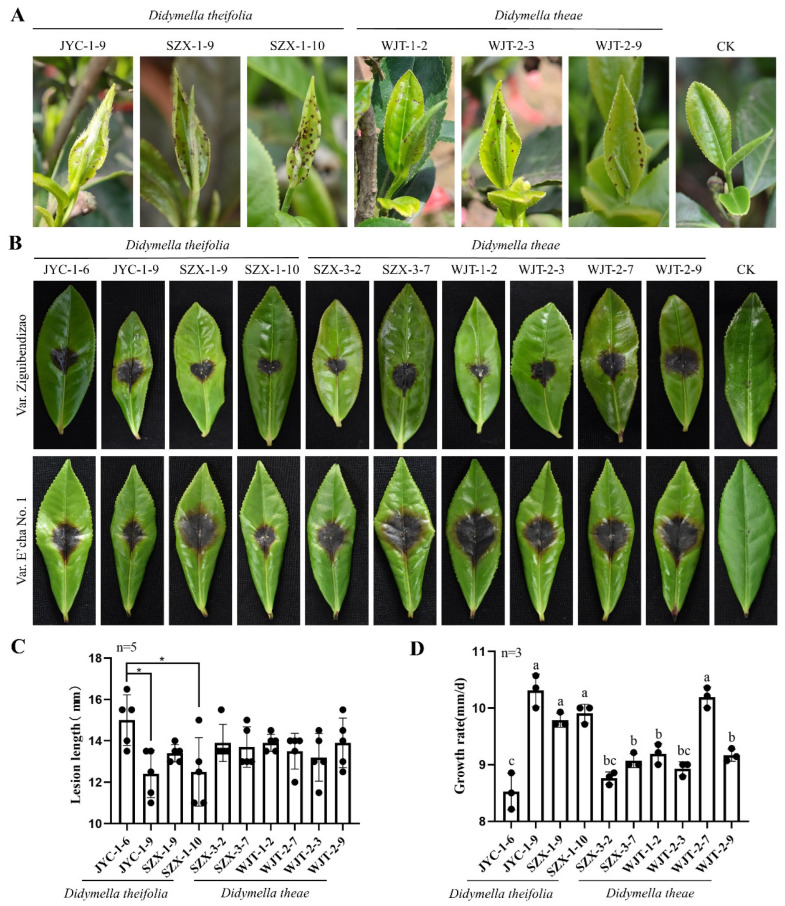
Virulence and pathogenicity tests and growth rates of *Didymella* isolates. (**A**) The representative leaf symptoms of the disease induced by the conidia solution of *Didymella* isolates (indicated on top of the panel) on the leaves of var. Enshitaizicha under natural field conditions. (**B**) The representative leaf symptoms of *Didymella* isolates on the leaves of vars. Ziguibendizao and E’cha No.1 at 4 dpi induced by mycelium discs under wounded conditions. (**C**) Bar graphs showing the virulence of *Didymella* isolates indicated by lesion lengths developed on tender leaves of tea (var E’cha No.1). (**D**) The growth rates of *Didymella* isolates in darkness at 25 °C for 7 days. The data were analyzed by SPSS Statistics 19.0 (International Business Machines Corporation, Armonk, NY, USA) using one-way ANOVA, and the mean values were compared with a Tukey test, *p* = 0.05. The letters and asterisks (*) on the error bars indicate the significant difference at *p* = 0.05.

**Table 1 jof-08-00782-t001:** Percentage of fungal colonies isolated from the sequenced samples.

Fungal Species	Colonial Isolation Percentage (%)			
	JYC	SZX	WJT	NC
*Didymella* sp.	20.0	21.1	20.0	0.0
*Pestalotiopsis* sp.	0.0	21.1	55.0	5.6
*Colletotrichum* sp.	20.0	21.1	0.0	22.2
*Diaporthe* sp.	10.0	5.3	0.0	11.1
*Alternaria* sp.	0.0	15.8	5.0	5.6
*Setophoma* sp.	0.0	5.3	0.0	0.0
*Arthrinium* sp.	0.0	0.0	10.0	0.0
*Pseudopithomyces* sp.	0.0	0.0	5.0	0.0
*Cercospora* sp.	10.0	0.0	0.0	0.0
*Melanconiella* sp.	10.0	0.0	0.0	0.0
*Phyllosticta* sp.	10.0	0.0	0.0	44.4
Unidentified	20.0	10.5	5.0	11.1

## Data Availability

Type specimens of *D. theae* and *D. theifolia* were deposited in HMAS under the accession numbers HMAS 351945 and HMAS 351946, and their ex-type living cultures were deposited in CGMCC under the accession numbers CGMCC 3.20886 and CGMCC 3.20887. Sequences generated in this study were deposited in GenBank under the accession numbers listed in Appendix A, and taxonomic information was deposited in MycoBank under the accession numbers MB 842446 and MB 842458.

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
