# Peer review of "Amplicon Sequencing Reveals Novel Fungal Species Responsible for a Controversial Tea Disease"

_jof, 2022, doi:10.3390/jof8080782_

Round 1

Reviewer 1 Report

This research used integrated approaches (amplicon sequencing, culturing, Koch postulates, phylogeny, morphology) to identify unknown fungal agents of a tea syndrome. Overall, I like their approach but here are some major comments:

1) The authors like to claim this research was the first one to do something (e.g. using amplicon sequencing in etiology). However, the merit of this study is really the “integration” of multiple methods. In fact, amplicon sequencing has been used in other studies to understand mycobiome of plant symptoms, but most of those studies did not complete follow up steps such as Koch postulates. If looking at the amplicon sequencing part of this research, there are several flaws. For example, no replicates were included and the materials and methods were unclear on how many leaves were included per sample. Because of these problems, no statistical analysis could be conducted. I suggest the author to reconsider how much emphasis to put on amplicon sequencing and highlight the integration part. 

2) The English writing need to be much improved. Some sentences are really long and the grammar was incorrect (some examples are listed below).

3) The overall writing can be improved. For example, some M&M was in introduction. Some really lengthy M&M was in the results. I suggest the authors to use supplementary tables for details of M&M (e. g. quality of read, details of collection site). The authors need to be more specific when referring to numbers (e. g. avoid using wordings like “over 100”, “considerably high”).

Some notes are below:

-Title: Is the word “amplicon sequencing” that important here? 

-page 1 Ln 17: “underdetermined”: do you mean “undetermined”? 

-page 1 Ln20: You really don’t need to write “informatics analysis”, unless more specific information is provided. 

- page 1 Ln 27: Try to avoid using too many “first”. For example: “This is the first report of Didymella fungi infecting tea plants besides D. segeticola and D. bellidis.” In the future, do you want people to claim some “first time” findings by keep adding species names to the “besides” list?

- The “Importance” section is pretty much the same as the abstract. If the “Importance” section is for a different group of readers, try to adjust it.

- “To our knowledge, this approach has not been applied in etiological identification of plant diseases before.” Amplicon sequencing has been widely used to understand possible fungi associated with plant disease before (e.g. Diaz-Cruz and Cassone 2020, Amplicon Sequencing Reveals Extensive Coinfections of Foliar Pathogens in Soybean).

-Page 2, Ln34: “It still cannot completely exclude other potential fungi responsible for LBSD since the previous identification strongly depended on the fungal isolation, which is easily omitted some pathogens if they are uncultured or difficult to grow on potato dextrose agar (PDA) medium.” Weird sentence structure. The “which … some pathogens” shouldn’t be passive.

-Page 2, Ln 37: “Moreover… of China”. I don’t really understand the logic of this sentence leading to the next sentence (“All of these call…”). Perhaps a bit more explanation is needed?

-Page 2, Ln 46-48: These should belong to the results, not the introduction.

-Page 2, Ln 51: Abbreviations (e.g. SZX) should be defined the first time they appear, not a few sentences later. A sampling map or a supplementary table will make this much easier to understand. Also, more details are required here. For example, how many leaves were included in each sample and how many replicates were collected?

-Section 2.2: “Uparse” to “UPARSE”; Mothur is a package, not an algorithm

-Section 2.3: How many leaf pieces per sample were tested?

-Section 2.6: “The sequences of obtained isolates were retrieved in NCBI (http://www.ncbi.nlm.nih.gov) database using BLASTn program, and the phylogenetically related fungal sequences were downloaded for further phylogenetic analysis.” It’s unclear what sequences (based on what criteria) were retrieved and if the search was comprehensive enough? Also, the English is confusing here.

-Section 3.1: Ln 13-17: Most of this can go to supplementary materials. 

-Section 3.1: Ln 19: “revealed over 100 fungal genera”: This is a scientific paper, please be specific.  

-Section 3.1: Ln 33: What value is “considerable high”? Also, should be “considerably high”.

-Section 3.1: Ln 39: Some of these should go to the Materials and Methods

-Section 3.4: Much of these (e.g. parameter settings, number of sites etc.) should be in the Materials and Methods

-Section 3.4: Can we learn something from the individual locus? Did the results from individual locus agree with each other? For example, by ITS alone, if we did BLAST or reconstruct a phylogeny, will we be able to identify these two fungi?

Author Response

Response to the reviewer

This research used integrated approaches (amplicon sequencing, culturing, Koch postulates, phylogeny, morphology) to identify unknown fungal agents of a tea syndrome. Overall, I like their approach but here are some major comments:

1) The authors like to claim this research was the first one to do something (e.g. using amplicon sequencing in etiology). However, the merit of this study is really the “integration” of multiple methods. In fact, amplicon sequencing has been used in other studies to understand mycobiome of plant symptoms, but most of those studies did not complete follow up steps such as Koch postulates. If looking at the amplicon sequencing part of this research, there are several flaws. For example, no replicates were included and the materials and methods were unclear on how many leaves were included per sample. Because of these problems, no statistical analysis could be conducted. I suggest the author to reconsider how much emphasis to put on amplicon sequencing and highlight the integration part. 

RESPONSE: We have changed our tone by removing some claim of the first report. Instead, we have enhanced to describe the integrated part.

2) The English writing need to be much improved. Some sentences are really long and the grammar was incorrect (some examples are listed below).

RESPONSE: We have asked Dr. Karim Shafik, from Alexandria University, Egypt, who has a good practice in English writing, severely improved the English grammar and sentence structures.

3) The overall writing can be improved. For example, some M&M was in introduction. Some really lengthy M&M was in the results. I suggest the authors to use supplementary tables for details of M&M (e. g. quality of read, details of collection site). The authors need to be more specific when referring to numbers (e. g. avoid using wordings like “over 100”, “considerably high”).

RESPONSE: We have checked and restructured the presentation according to the suggestions.

 Some notes are below:

-Title: Is the word “amplicon sequencing” that important here? 

RESPONSE: We tend to keep the word “amplicon sequencing” in the title considering that this approach has played an important role in this study. Moreover, we also try to emphasize the application of this protocol. 

-page 1 Ln 17: “underdetermined”: do you mean “undetermined”? 

RESPONSE: Yes, we have changed this word throughout the manuscript.

-page 1 Ln20: You really don’t need to write “informatics analysis”, unless more specific information is provided. 

RESPONSE: We removed some “informatics analysis”.

- page 1 Ln 27: Try to avoid using too many “first”. For example: “This is the first report of Didymella fungi infecting tea plants besides D. segeticola and D. bellidis.” In the future, do you want people to claim some “first time” findings by keep adding species names to the “besides” list?

RESPONSE: We removed some of these claims.

- The “Importance” section is pretty much the same as the abstract. If the “Importance” section is for a different group of readers, try to adjust it.

RESPONSE: Since the “Importance” section is not obligatory in a manuscript in journal JOF, we removed this section in the improved version.

- “To our knowledge, this approach has not been applied in etiological identification of plant diseases before.” Amplicon sequencing has been widely used to understand possible fungi associated with plant disease before (e.g. Diaz-Cruz and Cassone 2020, Amplicon Sequencing Reveals Extensive Coinfections of Foliar Pathogens in Soybean).

RESPONSE: We removed these claims.

-Page 2, Ln34: “It still cannot completely exclude other potential fungi responsible for LBSD since the previous identification strongly depended on the fungal isolation, which is easily omitted some pathogens if they are uncultured or difficult to grow on potato dextrose agar (PDA) medium.” Weird sentence structure. The “which … some pathogens” shouldn’t be passive.

RESPONSE: We have restructured the sentence as follows: “It still cannot completely exclude other potential fungi responsible for LBSD since the previous identification strongly depended on the fungal isolation, and some pathogens were easily omitted if they were uncultured or difficult to grow on potato dextrose agar (PDA) medium”

-Page 2, Ln 37: “Moreover… of China”. I don’t really understand the logic of this sentence leading to the next sentence (“All of these call…”). Perhaps a bit more explanation is needed?

RESPONSE: We have restructured the sentence as follows: “Moreover, Phoma segeticola var. camelliae was also reported in association with this disease in Guizhou province of China. Therefore, the LBSD etiology remains controversial and calls for a de novo identification of LBSD by a full-scale approach to get an insight of fungal pathogens inside the diseased plants.”

-Page 2, Ln 46-48: These should belong to the results, not the introduction.

RESPONSE: We have changed the presentation as follows: “Here we reveal Didymella spp. responsible for LBSD by ITS-based amplicon sequencing in combination with fungal isolation and fulfillment of Koch’s postulates. This study provides an integrated approach to identify the fungal etiology responsible for plant disease, and two novel Didymella species are proposed and characterized.”

-Page 2, Ln 51: Abbreviations (e.g. SZX) should be defined the first time they appear, not a few sentences later. A sampling map or a supplementary table will make this much easier to understand. Also, more details are required here. For example, how many leaves were included in each sample and how many replicates were collected?

RESPONSE: We have explained the abbreviation along with their first appearance. Due to only five samples, we intend to describe them directly rather than adding and a map or a supplementary table. We have added more necessary information about the samples here.

-Section 2.2: “Uparse” to “UPARSE”; Mothur is a package, not an algorithm

RESPONSE: We changed.

-Section 2.3: How many leaf pieces per sample were tested?

RESPONSE: At least four small pieces of tissues were excised from the each tea leaf, and this information has been involved.

-Section 2.6: “The sequences of obtained isolates were retrieved in NCBI (http://www.ncbi.nlm.nih.gov) database using BLASTn program, and the phylogenetically related fungal sequences were downloaded for further phylogenetic analysis.” It’s unclear what sequences (based on what criteria) were retrieved and if the search was comprehensive enough? Also, the English is confusing here.

RESPONSE: We have restructured the presentation as follows: “The obtained ITS sequences were aligned with the sequences deposited in NCBI (http://www.ncbi.nlm.nih.gov) database using the BLASTn program, and the phylogenetically related fungal isolates and their ITS, LSU, tub2, and rpb2 sequences were downloaded for further phylogenetic analysis”

-Section 3.1: Ln 13-17: Most of this can go to supplementary materials. 

RESPONSE: We have deleted some information since these information was already listed in Table S2.

-Section 3.1: Ln 19: “revealed over 100 fungal genera”: This is a scientific paper, please be specific.  

RESPONSE: We have replaced it with the specific number.

-Section 3.1: Ln 33: What value is “considerable high”? Also, should be “considerably high”.

RESPONSE: We proposed a value no less than 1% as a considerably high abundance here, and presented this information in the related sentence in the improved version, i.e., “Of these fungi, Didymella spp. were detected in all diseased samples with a considerably high abundance (no less than 1%, i.e., 5.27% in SZX, 4.32% in JYC, and 18.89% in WJT),”

-Section 3.1: Ln 39: Some of these should go to the Materials and Methods

RESPONSE: We did.

-Section 3.4: Much of these (e.g. parameter settings, number of sites etc.) should be in the Materials and Methods

RESPONSE: We did.

-Section 3.4: Can we learn something from the individual locus? Did the results from individual locus agree with each other? For example, by ITS alone, if we did BLAST or reconstruct a phylogeny, will we be able to identify these two fungi?

RESPONSE: We ever tried to construct the phylogenetic tree using the single locus ITS for all the species of this genus, and many species could not be distinguished including the both species obtained in this study. But the phylogenetic tree constructed with the locus rpb2 could distinguish most of them. Therefore, it is necessary to resolute these species with their multi-loci.

Reviewer 2 Report

Dear Authors,

Your manuscript was interesting and I started to read it carefully, once it comes to the new species identification, there is a big problem. The phylogenetic analysis of this paper is not acceptable.

Following the recent taxonomic treatments of this genus, there are over 20 species accepted, but you have only added 17. Did not follow the recent taxonomic treatments. Thus, your all observations need to make sure again. taxon sampling to run the tree is also poor, need to add at least 2 strains from each species. Identifying new species, required strong phylogenetic support with morphology. Try to follow RaxmL and Byasian analysis as your tree doesn’t have the proper topology it must have.  

Without confirming this problem I think it's not worth going further, as your study really depends on these two new species. therefore I will be not correct from this point forward.

Other specific comments are given in the file.

Author Response

Response to the reviewer

Dear Authors,

Your manuscript was interesting and I started to read it carefully, once it comes to the new species identification, there is a big problem. The phylogenetic analysis of this paper is not acceptable.

Following the recent taxonomic treatments of this genus, there are over 20 species accepted, but you have only added 17. Did not follow the recent taxonomic treatments. Thus, your all observations need to make sure again. taxon sampling to run the tree is also poor, need to add at least 2 strains from each species. Identifying new species, required strong phylogenetic support with morphology. Try to follow RaxmL and Byasian analysis as your tree doesn’t have the proper topology it must have.

Without confirming this problem I think it's not worth going further, as your study really depends on these two new species. Therefore I will be not correct from this point forward.

RESPONSE: We have collected all the species (i.e., 70 species) belonging to this genus, and reconstructed the phylogenetic tree using all these species (see Figure 3-origin below) by referring a recently reported taxonomic studies of this fungal genus References: Hou et al. The phoma-like dilemma. Stud Mycol 2020, 96, 309-396). However, as we included all these species in a phylogenetic tree, the tree size looks too big and the newly described species are not clearly indicated. Therefore, we have selected the 25 species that have the closest phylogenetic relationship with the newly obtained species as judged from the original phylogenetic tree constructed using all the species (Figure 3-origin), reconstructed the phylogenetic tree using Maximum Likelihood and Byasian analysis, and involved in our improved version. You can find that the taxonomic situation of each species in the improved Figure 3 keeps consistent with the original tree constructed with all species (see the both phylogenetic trees attached below). 

Figure 3. Phylogenetic tree generated by Bayesian inference (BI) and Maximum Likelihood (ML) based on combined ITS, LSU, rpb2 and tub2 sequence of Didymella theae, D. theifolia and the twenty species that have the closest phylogenetic relationship with them. Neoascochyta paspali (CBS 560.81) was selected as the outgroup. Bayesian posterior probability (PP≥ 0.90), and ML bootstrap support values (ML≥ 50%) are shown at the nodes (PP/ML).(Please find it on the attached file)

Figure 3-origin. Phylogenetic tree generated by Bayesian inference (BI) and Maximum Likelihood (ML) based on combined ITS, LSU, rpb2 and tub2 sequence of Didymella theae, D. theifolia and other 70 Didymella species. Neoascochyta paspali (CBS 560.81) was selected as the outgroup. Bayesian posterior probability (PP≥ 0.90), and ML bootstrap support values (ML≥ 50%) are shown at the nodes (PP/ML).(Please find it on the attached file)

Other specific comments are given in the file.

RESPONSE: We essentially agree with all comments and suggestions, we have improved the manuscript in the suggested orientation.

-Page 2, Ln 34-35, this section is almost similar as abstract.

RESPONSE: Since the “Importance” section is not obligatory in a manuscript in journal JOF, we removed this section in the improved version.

- Page 2, Ln 22, when was the 1st reported? only based on this you can mention "recent"

RESPONSE: We removed the word, and changed the presentation as follows: “Several years ago, a noticeable disease infecting tea plants, named tea leaf brown-black spot disease (LBSD) emerged in various tea-cultivated regions in China due to increased rainfall. ”

-Page 2, Ln 41-48, re-arrange the last sentence. it dons' t need to give a conclusion here. justify your study and mention objectives

RESPONSE: We have changed the presentation as follows: “Here we reveal Didymella spp. responsible for LBSD by ITS-based amplicon sequencing in combination with fungal isolation and fulfillment of Koch’s postulates. This study provides an integrated approach to identify the fungal etiology responsible for plant disease, and two novel Didymella species are proposed and characterized.”

- Page 2, Ln 26, 54, upper case

RESPONSE: We did.

-Page 3, Ln 4, no abbreviation when it starts a sentence

RESPONSE: We changed this situation.

-Page 3, Ln 6, it's better to mention how many samples from each site.

RESPONSE: We added this information.

-Page 3, Ln 16, lower case italic for all coding regions

RESPONSE: We did.

-Table 1, better be round off, 21.1

RESPONSE: We tend to keep these numbers in the same decimal places throughout the manuscript.

-Section 3.4, this have a serious problem need to be addressed. following recent taxonomic treatments of this genus, you need to conteict.

RESPONSE: As mentioned above, we have reconstructed the phylogenetic tree following recent taxonomic treatments of this genus. 

Round 2

Reviewer 1 Report

Fig. 1 & Fig. 3:

I totally agree with the other reviewer that three decimal points should be rounded up to 2 or 1 decimal point, and these details make the figure hard to read. In the response, the authors claimed they wanted to use the same decimal points throughout the paper. However, in Fig. 1, they have used three or two decimal points. 

Fig. 3: "Phylogenetic tree generated by Bayesian inference (BI) and Maximum Likelihood (ML)"": I understand you used Bayesian and ML approaches here. However, what is the tree topology based on? Did you overlay the bootstrap and posterior probability on a ML tree?

Page 4 Ln 32: "2.5. DNA extraction and determination of the taxonomic region sequences" This title doesn't make sense

Page 3 Ln7: I still recommend you provide a map, at least in the supplementary materials. This can be a simple map but let least the readers can know the distance among samples and the rough locations. This paragraph is hard to read for people unfamiliar with the names of these locations. 

page 4 Ln52

ITS=1-458 etc. please add the unit (e.g. bp).

Page 5 Ln 38-43:

Break the sentence to make it easier to read

"informatics analysis" is rarely used. You can try to google it and you can see this. Please change this to bioinformatic analysis or some other standard wordings. Please do the modification consistently throughout the manuscript.

Page 5 Ln 44-47: Break the sentence.

Page 5 Ln 50: This is a five-line sentence! Starting with "A phylogenetic tree of these Ascomycota fungi in the diverse taxonomic situation was constructed", the sentence needs to be rearranged. 

Page 6 Ln 11,12: "Although" and "while" should not appear in the same part of the sentence.

Page 8 Ln28: Replace "few" by "fewer"

Author Response

Response to Reviewer 1

Thank you very much for reading carefully our manuscript, making pertinent comments, and providing good suggestions. Since we essentially agree with all comments and suggestions, we have improved the manuscript according to the suggestion.

Reviewer 1.

Fig. 1 & Fig. 3: I totally agree with the other reviewer that three decimal points should be rounded up to 2 or 1 decimal point, and these details make the figure hard to read. In the response, the authors claimed they wanted to use the same decimal points throughout the paper. However, in Fig. 1, they have used three or two decimal points. 

RESPONSE: We change these numbers in the both figures with the same decimal points, i.e., two decimal points.

Fig. 3: "Phylogenetic tree generated by Bayesian inference (BI) and Maximum Likelihood (ML)"": I understand you used Bayesian and ML approaches here. However, what is the tree topology based on? Did you overlay the bootstrap and posterior probability on a ML tree?

RESPONSE: Here, we conducted the ML tree used to show the phylogenetic relationships, and overlaid the Bayesian posterior probability (PP≥ 0.90) and ML bootstrap values (ML≥ 50%) on the ML tree.

Page 4 Ln 32: "2.5. DNA extraction and determination of the taxonomic region sequences" This title doesn't make sense

RESPONSE: We have changed the description as “DNA extraction, amplification, and sequencing”.

Page 3 Ln7: I still recommend you provide a map, at least in the supplementary materials. This can be a simple map but let least the readers can know the distance among samples and the rough locations. This paragraph is hard to read for people unfamiliar with the names of these locations. 

RESPONSE: We have depicted a map of Hubei province, and marked the places where the samples were collected.

page 4 Ln52  ITS=1-458 etc. please add the unit (e.g. bp).

RESPONSE: We did.

Page 5 Ln 38-43: Break the sentence to make it easier to read

RESPONSE: We did.

"informatics analysis" is rarely used. You can try to google it and you can see this. Please change this to bioinformatic analysis or some other standard wordings. Please do the modification consistently throughout the manuscript.

RESPONSE: We have changed the word “informatics” with “bioinformatics” throughout the manuscript.

 Page 5 Ln 44-47: Break the sentence.

RESPONSE: We did.

Page 5 Ln 50: This is a five-line sentence! Starting with "A phylogenetic tree of these Ascomycota fungi in the diverse taxonomic situation was constructed", the sentence needs to be rearranged. 

RESPONSE: We have changed the sentence as “A phylogenetic tree constructed based on the ITS sequences of these Ascomycota fungi revealed 20 fungal species, belonging to 20 genera, 16 families, 10 orders, 7 classes of one this phylum, while their total amount was similar in each sample”

Page 6 Ln 11,12: "Although" and "while" should not appear in the same part of the sentence.

RESPONSE: We removed the word “Although”.

Page 8 Ln28: Replace "few" by "fewer"

RESPONSE: We did.

Reviewer 2 Report

Dear Authors

Sorry for the late response. You have revised the manuscript nicely. However, there are few minor changes are required as given in the PDF

Author Response

­­­­­­­­Response to Reviewer 2

Thank you very much for reading carefully our manuscript, making pertinent comments, and providing good suggestions. Since we essentially agree with all comments and suggestions, we have improved the manuscript according to the suggestion.

The section “Description” need to be standardized.

RESPONSE: We did as referring to the description Chen et al. Didymellaceae revisited. Studies in Mycology, 2017, 87: 105–159.

Where is Conidiophores conidiogenous cells? are there any clamidospore ?

RESPONSE: We have checked the conidiogenous cells, and involved their related information. We did not observed any clamidospore.

Page 16, Ln20, Remove the “spp.”

RESPONSE: We did